# A Historical Perspective on Dental Composite Restorative Materials

**DOI:** 10.3390/jfb15070173

**Published:** 2024-06-25

**Authors:** Jack L. Ferracane

**Affiliations:** Department of Oral Rehabilitation and Biosciences, Oregon Health & Science University, Portland, OR 97201, USA; ferracan@ohsu.edu; Tel.: +1-503-494-4327

**Keywords:** dental composite, fillers, monomers, light curing, clinical performance, physical properties, development, composition

## Abstract

This review article will discuss the origin of resin-based dental composite materials and their adoption as potentially useful adjuncts to the primary material used by most dentists for direct restorations. The evolution of the materials, largely driven by the industry’s response to the needs of dentists, has produced materials that are esthetic, strong, and versatile enough to be used in most areas of the oral cavity to replace or restore missing tooth structures. Significant advancements, such as the transition from chemical to light-curing materials, refinements in reinforcing particles to produce optimum polishing and wear resistance, formulating pastes with altered viscosities to create highly flowable and highly stiff materials, and creating materials with enhanced depth of cure to facilitate placement, will be highlighted. Future advancements will likely reflect the movement away from simply being a biocompatible material to one that is designed to produce some type of beneficial effect upon interaction within the oral environment. These new materials have been called “bioactive” by virtue of their potential effects on bacterial biofilms and their ability to promote mineralization of adjacent tooth structures.

## 1. Introduction

Resin-based composites (RBCs) for dental restorative materials have become the most used dental material due to their esthetics, properties, reasonable cost, versatility, and acceptable clinical performance. While there remains room for improvement based on less-than-ideal clinical longevity [1,2], there is no doubt that the initial development and further refinement of this material has revolutionized the practice of dentistry. Prior to the invention of dental composite as a restorative material, esthetic dentistry was restricted to the anterior portion of the mouth, first through the use of silicate cements, and followed in the early 1950s by polymethylmethacrylate (PMMA) resin consisting of a pre-polymerized PMMA powder mixed with methyl methacrylate (MMA) liquid monomer, in which curing was accomplished by an amine–peroxide reaction invented by German researchers in the 1940s [3]. These materials suffered from several deficiencies, including marginal staining and discoloration due to a high level of polymerization shrinkage, a lack of bonding to the tooth, and a huge mismatch in thermal expansion coefficient as compared with the tooth, as well as pulpal reactions due to the toxicity of MMA [4]. In addition, the material was inherently weak, limiting its indications [5,6,7]. Limited attempts were made to incorporate inorganic fillers with the powder to reduce the thermal expansion coefficient and reduce shrinkage and water sorption [8]. But clearly, improved materials were needed, and an attempt was made by Dr. Rafael Bowen to introduce epoxy resins filled with crushed fused quartz particles [9]. Epoxies had the ability to adhere to many surfaces, but they did not set well in moist conditions, and the chemicals available to cause rapid polymerization were relatively irritating to tissues. Further investigation by Dr. Bowen produced a hybrid molecule based on the reaction of a methacrylate with an epoxide, resulting in the Bis-GMA monomer (bisphenol A glycidyl methacrylate), often referred to as Bowen’s resin, for which a US patent (3,066,112) was issued in 1962 [10], and the material was introduced in the dental literature in 1963 [11]. This invention by Dr. Bowen paved the path for modern-day RBC materials, and for this he is owed a tremendous debt of gratitude by the profession. In comparison to the acrylics being used at the time, the new composites were stronger and stiffer, had less polymerization shrinkage, had a thermal expansion coefficient more similar to that of the tooth, and were more esthetic. Perhaps of equivalent importance was the popularization of the acid etch technique for enamel, attributed to Dr. Michael Buonocore, who introduced this as a mechanism to adhere the acrylic restorative to the tooth [12]. Acid etching with phosphoric acid provided the means by which the composite could mechanically adhere to and reliably seal the enamel margins of the restoration. Initially, the material was mainly limited to anterior restorations where esthetics were a primary concern, and most posterior direct restorations continued to be performed with dental amalgam. This would evolve over the years as further improvements were achieved in both the properties of the dental composites and their adhesion to the tooth (Figure 1).

There are numerous reviews of the composition and properties, as well as the clinical performance, of resin-based dental composites, and a few are cited here [1,13,14,15,16,17,18,19,20,21,22]. The purpose of this article is not to reiterate or to comprehensively assess the extensive literature in this field. Rather, the focus of this review article will be to chronologically identify some of the significant changes to the materials, specifically to the polymerization initiators, resin monomers, and fillers, that resulted in commercially available new products (Figure 2). It is important to note that many of the adjustments to the formulations of dental composites have been a direct response to the requests and needs of dental practitioners, often identified in focus group discussions with key opinion leaders. These efforts have led to the development of many excellent products that, when handled appropriately, can create excellent, long-lasting restorations, especially in patients who care for their oral hygiene. The article will close with a brief review of likely new directions for the continued development of dental composite materials.

## 2. Advances in the Process of Initiating Polymerization of Resin Composites

As noted above, the initial commercial dental composite materials were two-part paste systems, with one containing a peroxide initiator and the other an amine activator, with the pastes being mixed to cause polymerization in minutes through an auto- or self-curing process [23]. The major change that revolutionized the placement of composites came with the introduction of light-curing technology (see reviews by Stansbury [24], Ikemura and Endo [25], and Hadis et al. [26]). Ultraviolet light began to be used as an activator for sealants and composites containing benzoin methyl ether or similar UV-sensitive compounds around 1970, obviating the need for mixing two pastes and having to place the material within the confines of a defined working time. Despite showing acceptable results in a long-term clinical trial [27], due to limitations in depth of penetration of UV light (below 400 nm wavelength), as well as concerns over exposure to relatively high-intensity UV radiation, UV was replaced by visible light activation technology for curing dental composites in the late 1970s and early 1980s. Composites were made curable by visible light by incorporating camphorquinone (CQ) and a tertiary amine activator (co-initiator) within the single-paste resin-based sealant or composite. In this case, CQ is the light-sensitive molecule that most efficiently absorbs photons at around 468 nm in the blue region of the electromagnetic spectrum and is raised to a higher energy state. The amine then further reacts with the CQ, with the amine becoming the radical species that initiates polymerization through reaction with and splitting of the carbon–carbon double bond on the monomer. The activated monomer then reacts with another monomer and so on to propagate the polymer.

CQ is a very bright yellow compound, and failure to reduce it (i.e., bleach) during the polymerization process may lead to a lingering undesirable color in the material, especially an unfilled adhesive. Therefore, other photoinitiators of less intense color were sought to augment or replace CQ, such as 1-phenyl-1,2-propanedione (PPD), bisacylphosphine oxides (BAPO), triacylphosphine oxide (TPO), and the germanium-based Ivocerin [28,29]. These photoinitiators absorb light predominantly within the UV region and their range only extends into the visible region to around 400–410 nm. Many other types of photoinitiators have been developed for specific formulations of dental composites and adhesives [24]. For example, diphenyliodonium hexafluorophosphate has been combined with CQ to produce a system with enhanced efficiency and is commonly used in commercial products.

The first visible light-curing units contained quartz tungsten halogen (QTH) light bulbs emitting from below 400 to about 540 nm in the spectrum. These units incorporated filters to block out light emission outside of the 400–500 nm wavelengths to reduce potentially damaging UV output and excessive heat. Visible light curing provided an “on-demand” setting and deeper penetration of the activating light into the material than UV, thus enabling reliable curing of composites in increments of 2 mm, depending on the shade and translucency of the material. Theoretically, high-intensity blue light was considered safer than UV, though much concern continues to be raised over the exposure to blue light from dental and non-dental sources [30]. Because QTH bulbs are very inefficient in converting voltage to useful light energy, producing mostly heat, alternative light emission devices were sought. The major change in this regard was the introduction of light-emitting diodes (LEDs) to replace the QTH bulbs, thus reducing heat emission and eliminating the need for filters, as the LEDs typically emitted over the range of 420–500 nm (see the review of LEDs by Jandt and Mills [31]). The initial devices had low output, typically lower than that from the QTH lights. While LEDs were shown to be more efficient than QTH lights, their initial low outputs did not allow for sufficiently shorter curing times compared to the QTH lights. However, before long, LEDs with much greater power were produced, and eventually, by the mid-2000s, LED lights dominated the market and replaced QTH lights. Current LED curing lights typically have irradiance values of 800–2000 mW/cm^2^ [26]. Due to their lower power consumption requirements, these lights were more ideal than the QTH lights for use as battery-powered devices.

The evolution of light curing progressed with the introduction of lights emitting with greater power over the diameter of the light tip/guide, defined as irradiance (Watts/cm^2^), thus reducing the time needed to cure each increment to achieve the same level of carbon double bond conversion [32]. Initially, operators were typically exposing a single increment of composite for 40 s, but based on the concept of exposure reciprocity, where the most critical issue is the total energy received by the material (energy = irradiance × time, Joules/cm^2^), shorter curing times of 10–20 s became common as lights became more powerful. These times were reduced even further to 1 to 3 s for extremely high-powered lights (3000–4000 mW/cm^2^). The applicability of the theory of reciprocity to dental light curing has been discredited [33], and while the concept may apply to certain materials under certain circumstances [26,34], it cannot be utilized as a general rule but rather as guidance for increasing curing times when using lights with lower emissions. In the continued evolution of light curing, lights with high outputs were created in the 1990s by employing lasers and plasma arc (PAC) technology [32]. Most recently, an LED laser light-curing unit (Monet, AMD Lasers) has been commercialized for dentistry, demonstrating stable output over great distances with a well-collimated beam and operating at high irradiance [35]. It is important to note that there is considerable controversy in the field over the impact of using very high irradiance and short exposure durations on the quality and clinical performance of resin dental composites, and this area remains a focus of ongoing research.

Based on the need for light at appropriately low wavelengths, near the ultraviolet part of the spectrum, to stimulate the alternative “less yellow” photoinitiators, LED lights with chips having more than a single peak wavelength, otherwise known as multi-wave or poly-wave lights, were produced. These typically have two or three blue light-emitting chips and one near-violet-emitting chip (approximately 400–410 nm) to more efficiently cure composites with multiple photoinitiators [32]. In some cases, there are concerns with these lights in terms of a non-uniform distribution of light across the tip, as determined by profiling the spatial output of the emitted beam, which could result in reduced curing efficiency [36].

## 3. Advances in Monomer Technology in Dental Resin Composites

As noted in the Introduction, the dimethacrylate monomer, Bis-GMA, was used in the first successful dental composite materials, typically diluted with another dimethacrylate, such as triethyleneglycoldimethacrylate (TEGDMA). These monomers were made into pastes by the addition of ground quartz or glass particles, typically in the tens of micrometers but with many being around the size of a human hair at greater than 50 nm. The benefit of the Bis-GMA molecule over methylmethacrylate (MMA) was its much larger size (512 g/mol vs. 100 g/mol) and its ability to cross-link and form a thermoset polymer network due to the presence of the two carbon–carbon double bonds per molecule. Many different dimethacrylate monomers have been introduced in experimental systems as well as in commercial materials over the years [37,38,39], though there is no consistent evidence that any specific resin system has routinely outperformed any other from a clinical standpoint. While dental composite resins are generally believed to be biocompatible, concern has been raised over the release of bisphenol A (BPA) from the materials. Typically, BPA is not used in the formulation of composites, and while it is not a byproduct of the degradation of Bis-GMA, it may be formed by hydrolysis of other dimethacrylates [40]. Thus, many manufacturers use alternative monomers and tout their materials as being BPA-free.

One of the greatest challenges with resin dental composites has been overcoming the negative effects of polymerization shrinkage, which has been the topic of many studies and reviews [41,42,43,44]. Some of the most significant changes in the resin formulation for composites resulted from research addressing this important aspect. While the polymerization shrinkage of larger dimethacrylate monomers is several times less than that of smaller monomers, such as MMA, shrinkage during polymerization continued to be implicated in the lack of a tight seal between the restorative and the tooth surface, leading to microleakage. In time, it was determined that the shrinkage itself was not necessarily the main reason for the generation of marginal defects and interfacial gaps in composite restorations, but rather the internal stress created by the interaction between the polymerization shrinkage of the resin being resisted by the resin adhesion to the cavity preparation [45]. The most direct approach to resolving this problem was to create monomers with less polymerization shrinkage [46]. Typically, reduced shrinkage was achieved through the incorporation of still larger/longer monomers, reducing the amount of dimensional change attributed to each monomer-to-monomer connection [47]. These efforts arising from the industry and academia resulted in a number of commercial composite products incorporating dimethacrylate monomers, with names such as DX511 (Kalore, GC, Tokyo, Japan), TCD-urethane (Venus Diamond, Kulzer), and dimer dicarbamate dimethacrylate (N’Durance, Septodont) [48,49].

Another approach pursued to reduce shrinkage was based on the curing of monomers with epoxide functional groups, such as oxiranes or oxaspiro molecules [50]. This work recalled Bowen’s original idea of using epoxies based on their low curing shrinkage and good mechanical properties [9]. The commercial product arising from this work was a composite based on a multifunctional oxirane molecule (Filtek LS Silorane, 3M ESPE) polymerized by a cationic ring-opening reaction rather than a free-radical methacrylate polymerization [51]. Because of the different polymerization mechanism, this material was not compatible with typical adhesives and required its own dedicated product. The concept behind this system to show reduced curing shrinkage is that ring opening involves some expansion that offsets the contraction as the new covalent bond is formed and two molecules move closer together, reducing the overall free volume in the system. This new material showed good physical properties, but despite the significantly reduced curing shrinkage compared to conventional dimethacrylates, clinical studies showed equivalent but not significantly improved performance [52,53].

Another approach to using an alternative matrix system for dental composites was commercialized in the later 1990s and became a new category called ormocer (organically modified ceramic). These materials are hybrids of inorganic and organic molecules, in which, through a sol–gel process, a molecule with a long inorganic silica backbone is produced with organic side chains of polymerizable methacrylates [22]. While ormocers are cured by conventional light-curing techniques, due to their large size, they demonstrate lower polymerization shrinkage than conventional composites. The initial products (Admira, Voco GmbH; Definite, Degussa AG; and Ceram-X, Dentsply) that were brought to market also contained traditional dimethacrylate diluents to alter viscosity. These materials have undergone extensive in vitro and clinical evaluation. Systematic reviews of clinical studies showed that the ormocer restorations performed similarly to bulk-fill composites but were slightly less successful than conventional composites in posterior restorations [54] and tended to show greater tooth sensitivity and marginal degradation, perhaps due to the inclusion of dimethacrylate diluents in the formulations [55].

Another direction for dental composites was to create a material capable of adhesion to the tooth structure without requiring a separate application of a resin-based adhesive. The first materials were created as flowable dental composites based primarily on the same chemistry used in current dental adhesives. The materials experienced mixed success, with most in vitro and clinical studies showing that a traditional flowable composite applied with a traditional adhesive had equal or better overall performance [56,57]. Subsequently, manufacturers attempted to develop more heavily filled direct restoratives with self-adhesive potential. These tended to mimic, to some extent, glass ionomer- or compomer-type materials relying on carboxylic acid functional groups to provide adhesion to enamel and dentin. One material (Activa, Pulpdent) designed to be used without a separate adhesive showed poor clinical performance in a study within one year, and the study was discontinued [58]. However, when used with an adhesive in primary teeth in another clinical study, the material showed an acceptable performance [59]. At least one other self-adhesive bulk-fill material (SureFil One, Dentsply) emerged, showing reasonable success [60,61,62], but this product was discontinued due to supply issues that made it impossible for the manufacturer to continue to deliver the original product.

A major advancement was achieved in the early 2010s with the development of the bulk-fill composites, designed to be placed in increments of 4–5 mm and cured with a single light exposure. While the demand for improved light transmission through the material to achieve enhanced depth of cure was paramount for these materials, another critical requirement was to specifically formulate a material that produced low polymerization stress during curing. This was required because previous studies had shown that shrinkage stress and microleakage were correlated with resin volume [63,64]. The first materials were designed as flowable, such as SDR Flow (Dentsply), to achieve good adaptability to the cavity preparation and easy placement. However, due to the slightly reduced filler volumes required to produce their flowable nature, these materials tended to have lower mechanical properties and were designed to be used with a “capping” layer of a typical posterior composite applied as the occlusal surface. In time, other formulations were developed that could be used to fill the entire cavity preparation, though to a maximum of 4–5 mm in depth, depending upon the material. Enhanced cure depth can be accomplished by matching the refractive index of the reinforcing filler to that of the matrix resin, as well as by reducing light scattering within the composite by the incorporation of larger-sized filler particles. Both approaches allow enhanced light penetration to the depths of the restoration. In addition, at least one material (Tetric Evoceram Bulk Fill, Ivoclar) incorporated a more efficient photoinitiator for enhanced promotion of polymerization at lower light energy [36]. In addition, interesting monomer systems were developed that provided internal molecular reorganization during curing to reduce the polymerization contraction stress; commercialized examples (Filtek One Bulk Fill, 3M; Tetric PowerFill, Ivoclar) incorporated covalent adaptable networks involving addition–fragmentation chain transfer reactions [65,66]. In general, in vitro and clinical assessments of the bulk-fill composites have shown them to perform as well as conventional composites placed incrementally [67,68].

Other efforts to produce dental composites with lower polymerization stress have included the incorporation of nanogels and thiourethane oligomers. Nanogels are globular particles on the nanoscale made from internally cross-linked polymers that have been developed for a variety of uses, including drug delivery. These particles can also be incorporated into the resin matrix of composites [69] or chemically attached to the filler particles [70] to reduce contraction stress development during curing. Thiourethane oligomers can also be included directly in the resin matrix [71] or chemically attached to filler particles [72] to reduce composite stress by slowing the polymerization reaction, with no negative effect on the degree of conversion and actually producing a substantial increase in fracture toughness.

## 4. Advances in Filler Technology in Dental Resin Composites

Originally, dental composites contained relatively large quartz or glass particle fillers as the reinforcing agents. These particles were excellent strengtheners as the size of the particle enabled extensive crack blunting and deflection mechanisms. However, composites with average particle sizes approximating 10–20 µm, with the largest particles exceeding 50 µm, suffered from several drawbacks, which were evident in their poor wear characteristics and difficulty in producing and maintaining highly polished surfaces [73]. Wear proceeded in an uneven manner, beginning with the loss of the softer resin between the harder particles, which caused the latter to be loosened and released from the surface to create large voids and a rougher surface prone to further wear [74]. The solution to the problem was to create composites with reinforcing fillers of smaller sizes. Perhaps the most significant advancement of the time was the production and use of aerosol silica nanofillers with average sizes of approximately 40–50 nm, which were originally called “microfillers”, an inaccurate term but likely one that was more understandable at the time. So-called “microfill” composites became very popular due to their ability to produce and maintain a high gloss in the mouth because the particles were so small that they wore at a rate similar to the adjacent resin matrix, leaving the surface smooth [75]. Microfill composites tended to show superior wear against abrasive forces, such as toothbrushing, but tended to lose material at a faster rate than conventional composites when exposed to heavy occlusal forces [76]. The wear resistance of composites was shown to be proportional to certain mechanical properties, such as flexure strength and fracture toughness, both of which are a function of the amount of reinforcing fillers present [77]. Due to their small size, filler loading of microfills was limited to below 50% by volume, thus also limiting physical properties. To maximize filler content, manufacturers produced agglomerates of higher concentrations of particles in resin that were pre-polymerized as blocks and then ground into particles 50 µm or larger in size. Because the size of the individual hard inorganic particles remained on the nanoscale, these microfill composites had excellent polishing characteristics [78].

The next attempt to improve the materials was to adopt the concept of the “hybrid” composite. By coupling the larger reinforcing fillers with the smaller nanoparticles, the nanoparticles would fill the spaces between the larger particles and maximize filler loading and properties, such as wear resistance [74]. Theoretical studies have shown that ideal physical properties could be achieved by considering the packing of fillers of various size ranges [79,80]. However, the relatively large size of the particles in the original hybrid composites continued to be an issue for wear, leading to enhanced processes for grinding the glass particles to smaller sizes for mixing with the nanoparticles. This resulted in the introduction of new hybrid formulations called minifills [81], which later became known as microhybrids [14]. In some cases, manufacturers also incorporated some of the same types of pre-polymerized resin fillers they incorporated into the microfill composites, referring to these materials as nanohybrids [14]. Aside from the original “microfills”, there has been only one commercial dental composite (Filtek Supreme, 3M) composed solely of nanosized particles in high concentration, i.e., a true nanofill [82]. This material exhibited good properties, excellent wear resistance, and polishability. Other formulations, such as Estelite Sigma and Omnichroma (Tokuyama), are called supra-nanofills as they have spherical particles of uniform size in the 100–300 nm range and show excellent polishability [83]. Omnichroma, Transcend (Ultradent), Admira Fusion x-tra (Voco), and Venus Diamond/Pearl One (Kulzer) are some of the first dental composites designed as single-shade materials to be used in any tooth by virtue of their ability to blend in with their surroundings. Other materials limited to only one or a couple of shades have followed as a result of a general trend to simplify commercial composite systems.

Another significant change in dental composite technology was the introduction of both flowable and packable dental composites. The former were produced with either a lesser amount of the typical glass reinforcing particles to make the paste more fluid but also less strong or by specifically formulating resins with reduced viscosity to allow for relatively high filler loading to produce better physical properties [84]. In contrast, packable composites have much higher viscosity and stiffness during placement, and this is achieved primarily by altering the filler particle distribution and not the overall filler amount, as evidenced by the fact that these materials typically possess similar properties as conventional composites [85]. One packable material (Sonicfill, Kerr) was formulated to be dispensed with sonic energy through a special handpiece to reduce viscosity and enhance flow and adaptation to the cavity preparation.

It would be an understatement to state that many different types of inorganic compounds have been utilized or proposed as fillers for dental composites, and this topic has been included in several reviews [86,87]. Originally, fillers were composed of radiolucent silica or quartz, making it difficult to detect recurrent caries around dental restorations. The introduction and use of a variety of heavy-metal-containing glasses based on barium, aluminum, strontium, and zirconium, among others, produced radiopaque fillers with improved diagnostic advantages. The physical properties of a dental composite are largely affected by the filler content, and the effect of filler size, distribution, shape, and amount on the physical properties and degree of cure has been discussed in detail [86]. Based on the concept of the rule of mixtures, the approach typically has been to maximize the amount of filler present in the composite, except for flowable materials, to optimize the mechanical properties. This is typically achieved by incorporating a mixture of particles to enhance particle packing and minimize interparticle spacing, a critical factor for wear resistance. While a few dental composites have relied on spherical fillers, as mentioned above, most incorporate particles of irregular shape produced by a milling/grinding process. The exception to this is the inclusion of nano- or micro-sized fibers as reinforcement for the resin matrix. Fiber reinforcement has long been used in the plastics industry. Evidence shows that fiber reinforcement of dental composites can produce materials, such as Alert (Pentron), with elevated fracture toughness and strength [85]. Initial attempts produced composites that had poor polishability and wear resistance due to the fibers being exposed on the surface and subsequently breaking off and leaving defects within the material [88]. Other efforts successfully produced stronger, stiffer, and tougher composites through the addition of ceramic whiskers, such as silicon nitride [89], but these materials needed to be heat-cured as they were not amenable to light curing due to their opaque nature. More recent attempts to incorporate fibers into dental composites have produced other commercial products, such as everX Posterior (GC), which is a bulk-fill composite designed for use with a non-fiber reinforced composite as a capping layer.

## 5. New Directions in Dental Composite Technology

The future development of dental composite technology will most certainly be focused on the concept of “bioactivity”, a term that is used frequently by dental manufacturers and researchers but without a true consensus on its definition. The concept suggests that the material does not simply serve its primary purpose of restoring the tooth to function by replacing lost components of the tooth but that it produces some additional beneficial effect. By the broad definition of the term, this effect could be achieved by chemical processes, biological processes, or some combination thereof [90]. For dental composites, the vision for “bioactive” composites includes materials that can interact with the microbial environment to affect the recurrence of the caries process or materials that can reconstitute tooth structure that has been lost or damaged by the caries process, i.e., what is commonly referred to as remineralization. There have been many excellent reviews of the potential materials that may be employed in dental composites to impart “bioactivity” [91,92,93].

The possible approaches to combating oral biofilms involve the production of materials that are anti-fouling, keeping proteins and/or bacteria from adhering to their surface; antimicrobial, in which bacteria are actually killed by the material either through direct contact or through the release of a toxic component from the composite; or anti-biofilm, through the release of components that do not kill bacteria, but affect the process of forming a viable, adherent biofilm (Figure 3) [94]. Currently, there is a dental adhesive, Clearfil Protect Bond (Kuraray), that contains a tethered (i.e., non-releasing) quaternary ammonium compound, 12-Methacryloyloxydodecyl pyridinium bromide (MDPB), which causes bacterial cell death on contact and has shown some ability to reduce mineral loss of adjacent dentin in an in situ model [95], but has not shown improved clinical performance when compared to an adhesive that did not contain the MDPB monomer [96]. Another composite, Infinix (Nobio), is also designed with a quaternary ammonium molecule tethered to a silica nanoparticle filler and has shown reduced mineral loss in an in situ gap model [97]. Numerous composites have been developed with ion-releasing fillers (Activa, Pulpdent; Beautiful, Shofu) intended to aid in the remineralization of adjacent tooth structures and possibly reduce the loss of minerals (Figure 3). Recently, another ion-releasing composite, Cention N, based on what is referred to as an “alkasite” filler of calcium fluorosilicate, was shown to perform acceptably in posterior restorations over two years [61]. The full evaluation of the effectiveness of these “bioactive” materials will only be determined through extensive clinical research in years to come. It should also be noted that depending upon the claims made by the manufacturers for such materials, in order to demonstrate safety and effectiveness for regulatory agencies, such as the Food and Drug Administration in the United States, a less restrictive 510K process (demonstrating substantial equivalence to existing products) or a far more extensive and expensive premarket approval process would be required.

Another area that is of intense interest for development is that of self-healing composites. Materials can be designed to stop cracks from propagating to failure by healing the damaged crack region using either an intrinsic (i.e., incorporation of encapsulated healing agents in the material that are released during cracking) or extrinsic (i.e., application of external sources of energy such as heat or light) approach [98,99]. Such materials would be capable of responding to the stress field created by a propagated crack to repair and close it, making it more resistant to further cracking. Currently, no such product exists in dentistry, but it is an area of active research [100,101].

## Figures and Tables

**Figure 1 jfb-15-00173-f001:**
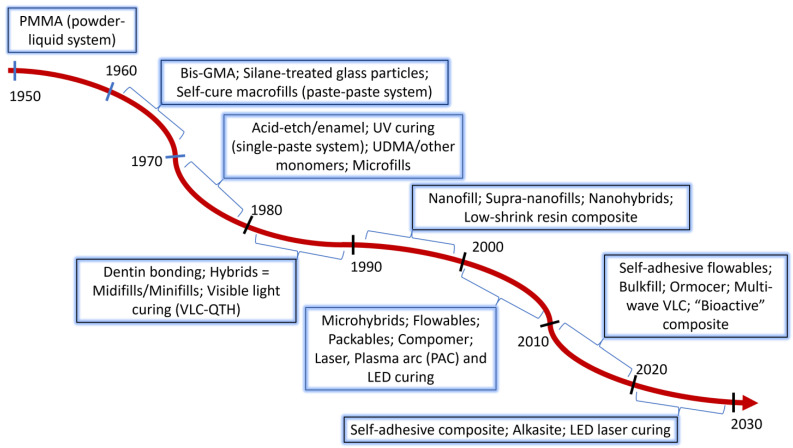
Approximate timeline for the development of dental composite restorative systems.

**Figure 2 jfb-15-00173-f002:**
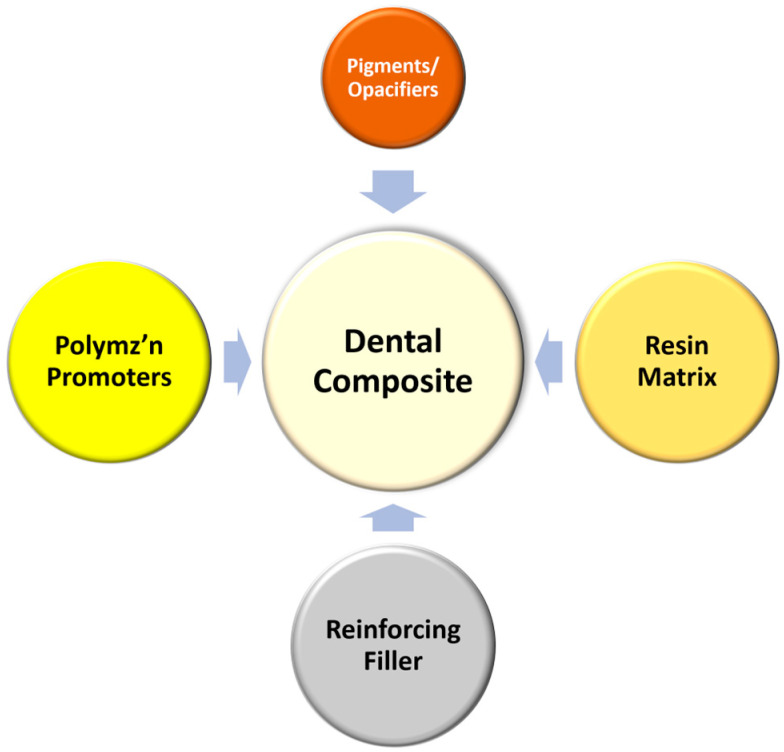
The basic components of dental composite materials.

**Figure 3 jfb-15-00173-f003:**
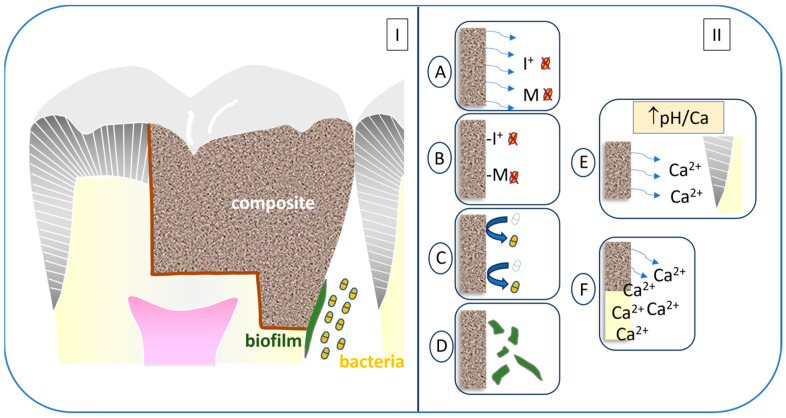
**(I**) Composite in a tooth with bacteria/biofilm condition. (**II**) (**A**) Ion/molecule release for bacterial killing. (**B**) Tethered ion/molecule for contact killing. (**C**) Anti-fouling surface. (**D**) Biofilm-disruptive surface. (**E**) Ion release–buffering. (**F**) Remineralizing.

## Data Availability

This article did not involve the creation of new data.

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
