# Peer review of "A Historical Perspective on Dental Composite Restorative Materials"

_jfb, 2024, doi:10.3390/jfb15070173_

Round 1

Reviewer 1 Report

Comments and Suggestions for Authors

This is an excellent review article presenting an informative historical narrative of the invention and development of resin-composites as direct restorative materials in dentistry.

The text presented is very well written and accurate and as such I have no corrections (despite reading it thoroughly).

My suggestions are to explore adding and or expanding on a couple of things. I understand that you could keep on expanding and can easily be turned into a book chapter or even a book.

Therefore, my suggestions are exactly that and I'm happy to leave it to the discretion of the author and editor to proceed accordingly.

Suggested areas:

In light curing you can consider adding 1-2 sentences on the latest ultra-fast polymerisation (suggested references below).

A. Ultra-fast photo-polymerisation

1. Ilie And Watts, Dent Mater. 2020 Apr;36(4):570-579.

2. Thanoon et al., Dent Mater. 2024 Mar;40(3):546-556.

There is a limited mention of photo-initiators and this could be slightly expanded to highlight new photo-initiators like Ivocerin that are employed in some of the new bulk-fill materials.

In terms of fillers, in some experimental formulations, new fillers like graphene, graphene nanoplatelets etc., are utilised.

Finally, for self-healing (I appreciate it is only a quick mention at the end), but a couole more up-to-date references can be used as well (suggestions below).

B. Self-healing composites

1. Yao et al., Dent Mater. 2023 Nov;39(11):1040-1050.

2. Althaqafi et al., J Funct Biomater. 2022 Feb 14;13(1):19.

Author Response

Comment 1: This is an excellent review article presenting an informative historical narrative of the invention and development of resin-composites as direct restorative materials in dentistry.

The text presented is very well written and accurate and as such I have no corrections (despite reading it thoroughly).

My suggestions are to explore adding and or expanding on a couple of things. I understand that you could keep on expanding and can easily be turned into a book chapter or even a book.

Therefore, my suggestions are exactly that and I'm happy to leave it to the discretion of the author and editor to proceed accordingly.

Response: Thank you for the kind comments about the manuscript. I have tried to respond to the suggestions to expand certain areas, being cautious about the overall length of the manuscript.

Comment 2: In light curing you can consider adding 1-2 sentences on the latest ultra-fast polymerisation (suggested references below).

  1. Ultra-fast photo-polymerisation
  2. Ilie and Watts, Dent Mater. 2020 Apr;36(4):570-579.
  3. Thanoon et al., Dent Mater. 2024 Mar;40(3):546-556.

Response: Thank you for the suggestion. I think that the concept of rapid cure with short exposures was covered in this paragraph (lines 143-145). However, the suggestion to include the citation of Illie and Watts is excellent, as they also describe a second composite using RAFT technology, PowerFill (Ivoclar). This material and the citation have now been included in line 271.

Comment 3: There is a limited mention of photo-initiators and this could be slightly expanded to highlight new photo-initiators like Ivocerin that are employed in some of the new bulk-fill materials.

Response: Ivocerin was noted in line 110 as a germanium-based PI.

Comment 4: In terms of fillers, in some experimental formulations, new fillers like graphene, graphene nanoplatelets etc., are utilised.

Response: As you alluded in your initial comment, we could fill a book simply describing all the different fillers that have been tried in dental composites. I tried for the most part to focus not on experimental materials, but on the materials/additives that led to commercial products. Therefore, I have not included many of these different innovations. I did not make any changes.

Comment 5: Finally, for self-healing (I appreciate it is only a quick mention at the end), but a couple more up-to-date references can be used as well (suggestions below).

  1. Self-healing composites
  2. Yao et al., Dent Mater. 2023 Nov;39(11):1040-1050.
  3. Althaqafi et al., J Funct Biomater. 2022 Feb 14;13(1):19.

Response: Thank you for the suggestion. I have added the Yao et al., citation at line 428.

Reviewer 2 Report

Comments and Suggestions for Authors

The author presented the Historic Perspective on Dental Composite Restorative Materials. Overall, the review is nicely written, and the author covered quite few aspects of the dental composite. However, there are few comments that should be addressed in this review article.

1- Toxicity related to bisphenol-A and how this issue can be addressed?

2- More in-depth information related to bioactive materials (compounds) is required. The bioactive materials (mainly hydroxyapatite, amorphous calcium phosphate, etc.) have been incorporating for the last three decades, however, still a commercial composite is not in the market, and if there, not widely available. 

3- Few comments related to new material such as Cention-N, that would be interesting for the readers and why Activa Bioactive could not perform well clinically?

Author Response

The author presented the Historic Perspective on Dental Composite Restorative Materials. Overall, the review is nicely written, and the author covered quite few aspects of the dental composite. However, there are few comments that should be addressed in this review article.

Comment 1: Toxicity related to bisphenol-A and how this issue can be addressed?

Response: Thank you for the suggestion. I do not believe that BPA is a significant issue with dental composites, as concluded in the article by Schmalz and Widbiller. BPA has been found in very few dental materials, and typically these were sealants. BPA is not used in the synthesis of dental composites, though it has been suggested to be a potential contaminant leftover from the synthesis of a precursor molecule for Bis-GMA. However, BPA can be formed from hydrolysis of bis-DMA which has been used in some commercial materials in the past. I have added the sentences below, as well as the Schmalz citation [40], at lines 178-183.  

While dental composite resins are generally believed to be biocompatible, concern has been raised over the release of bisphenol A (BPA) from the materials. Typically, BPA is not used in the formulation of composites, and while it is not a byproduct of the degradation of Bis-GMA it may be formed by hydrolysis of other dimethacrylates [40]. Thus, many manufacturers use alternative monomers and tout their materials as being BPA-free.

  1. Schmalz G, Widbiller M. Biocompatibility of Amalgam vs Composite - A Review. Oral Health Prev Dent. 2022 Mar 14;20(1):149-156.

Comment 2- More in-depth information related to bioactive materials (compounds) is required. The bioactive materials (mainly hydroxyapatite, amorphous calcium phosphate, etc.) have been incorporating for the last three decades, however, still a commercial composite is not in the market, and if there, not widely available.

Response: As the reviewer correctly notes, while there has been and continues to be intense research into so-called bioactive composite restorative materials, few commercial products exist, except for those already mentioned in the second to last paragraph of the article. Dozens of compounds have been incorporated into composites for this purpose, but these materials remain experimental and therefore were not included in this historical perspective. Because there is no proven clinical effect from these materials, it is not currently possible to further evaluate them. Therefore, no changes were made in the article.

Comment 3- Few comments related to new material such as Cention-N, that would be interesting for the readers and why Activa Bioactive could not perform well clinically?

Response: Regarding Activa, I believe that the poor clinical outcome in one study was because it was not used with an adhesive, as the manufacturer stated it could be. I stated the following in the article “One material (Activa, Pulpdent) designed to be used without a separate adhesive showed poor clinical performance in a study within one year, and the study was discontinued [56]. However, when used with an adhesive in primary teeth in another clinical study, the material showed an acceptable performance [57].”

Regarding Cention N, I have now included a reference to it in the text at line 407-409 along with a recent clinical study in which it demonstrated acceptable clinical performance [98].

Recently, another ion releasing composite, Cention N, based on what is referred to as an “alkasite” filler of calcium fluorosilicate, was shown to perform acceptably in posterior restorations over two years [98].

  1. Albelasy EH, Hamama HH, Chew HP, Montasser M, Mahmoud SH. Clinical performance of two ion-releasing bulk-fill composites in class I and class II restorations: A two-year evaluation. J Esthet Restor Dent. 2024 May;36(5):723-736.

Reviewer 3 Report

Comments and Suggestions for Authors

This is a well-written brief historical review with intensive future perspectives of dental composite filling materials. A few minor points may improve the readability and add value as a review article:

  1. L50: The patent can be included in the reference list with its title and date information.
  2. L326-327: The statement “Omnichroma is … Other materials limited to …” seems to promote a commercial brand instead of a category of materials. This should be supported with solid scientific references.
  3. L390: The full name of MDPB should be spelled out.
  4. L396: List the core techniques for the products Activa and Beautiful.

Additionally, I suggest including the following points to enhance the review:

  1. A brief coverage of the regulatory perspective on grandfathered Bis-GMA and methacrylate monomers, and the development of future new monomers in restorative dental composites, including functional monomers with antibiofilm, antimicrobial, and antifouling features.
  2. Mention expanding new dental composite materials to CAD/CAM millable and printable restorative materials.

Author Response

This is a well-written brief historical review with intensive future perspectives of dental composite filling materials. A few minor points may improve the readability and add value as a review article:

Comment 1: L50: The patent can be included in the reference list with its title and date information.

Response: Now included in the reference list as citation 10.

Comment 2: L326-327: The statement “Omnichroma is … Other materials limited to …” seems to promote a commercial brand instead of a category of materials. This should be supported with solid scientific references.

Response: No intent was made to promote Omnichroma, but it was the first single shade composite and has the longest history. To reduce the concern over propriety, I have now included other single shade composites in this same sentence, which now reads as follows (lines 333-334):

Omnichroma, Transcend (Ultradent), Admira Fusion x-tra (Voco) and Venus Diamond/Pearl One (Kulzer) are some of the first dental composites designed as single shade materials to be used in any tooth by virtue of their ability to blend in with their surroundings.

Comment 3: L390: The full name of MDPB should be spelled out.

Response: This has been included, 12-Methacryloyloxydodecyl pyridinium bromide. (line 398)

Comment 4: L396: List the core techniques for the products Activa and Beautiful.

Response: I am not sure what the reviewer is asking for here. In any case, as these are composites they may be used in all applications that a conventional material would be used.

Comment 5: Additionally, I suggest including the following points to enhance the review: A brief coverage of the regulatory perspective on grandfathered Bis-GMA and methacrylate monomers, and the development of future new monomers in restorative dental composites, including functional monomers with antibiofilm, antimicrobial, and antifouling features.

Response: This is a large subject, but I have included a sentence referring to the regulatory requirements for materials that make claims of “bioactivity” on lines 411-416.

“It should also be noted that depending upon the claims made by the manufacturers for such materials, in order to demonstrate safety and effectiveness for regulatory agencies, such as the Food and Drug Administration in the United States, a less restrictive 510K process (demonstrating substantial equivalence to existing products) or a far more extensive and expensive Premarket Approval process would be required.”

Comment 6: Mention expanding new dental composite materials to CAD/CAM millable and printable restorative materials.

Response: This article, for the purpose of brevity, did not include a discussion of indirect materials, either self-cured, heat-cured, CAD-CAM, or 3D printed. This would all be beyond the scope of this work, so no changes were made.